

# Multi-angle perception and convolutional neural network for service quality evaluation of cross-border e-commerce logistics enterprise

ShuTong Zhao[1], Zhenjie Yin[2] and Pingping Xie[1]

[1] College of Management, Suqian University, Suqian, Jiangsu, China
[2] Guangxi Xinsha Engineering Consulting Co., Ltd, Nanning, China

## ABSTRACT

The development of cross-border e-commerce logistics services has injected new vitality into the development of international trade, and therefore has become a new hot spot in theoretical research. In order to ensure the healthy development of cross-border e-commerce, it is urgent to build a set of scientific and effective evaluation mechanisms to scientifically evaluate the logistics service quality of cross-border e-commerce. Multi-angle perceptual convolutional neural network is a framework for service scene identification of cross-border e-commerce logistics enterprises based on deep convolutional neural network and multi-angle perceptual width learning. In this article, both shallow features and deep features were input into the deep perception model (DPM) to obtain a set of distinguishable features with causal structure, which was used to completely describe the high-level semantic information of cross-border e-commerce logistics enterprise services. Among them, DPM mainly adopts the fusion strategy of shallow feature and deep feature. Meanwhile, the feature representation is input into the width learning pattern recognition system for training and classification, so as to evaluate the service quality of cross-border e-commerce logistics enterprises. The multi-angle perceptual convolutional neural network can effectively solve the problems of high similarity between service classes of cross-border e-commerce logistics enterprises and large differences within the class, and achieve better generalization performance and algorithm complexity than support vector machine, random forest and convolutional neural network.

# INTRODUCTION

At present, more and more enterprises have changed from the original price competition to the current service quality competition (*Feng & Chen, 2022*). Service quality is the key to customer perception. When there is little difference in price, customers are more inclined to choose good service quality goods (*Yang & Lin, 2022*). Therefore, improving service quality has become an important means for cross-border e-commerce enterprises to win customers in the fierce market competition. There are a large number of cross-border

Corresponding author
Zhenjie Yin, zhenjieyin68@163.com

e-commerce enterprises in China, and their service quality is uneven. Slow delivery, goods lost, slow logistics, customer service response time is too long and other phenomena are widespread, the quality of service problems need to be solved. Therefore, it is necessary to build a service quality evaluation system for export cross-border e-commerce enterprises, so as to improve service quality in a targeted way, protect the legitimate rights and interests of consumers, and win competitive advantages for enterprises (*Wen, 2022*; *Li & Zhang, 2021*).

Deep learning technology has been widely concerned by researchers for its efficient representation of large-scale data. As one of the important methods, convolutional neural network has been widely applied from simple scene recognition to complex classification analysis (*Lakshmi et al., 2020*). The structure design and superparameter selection of convolutional neural network are the main factors that determine its performance and are also the focus of research in various applications. According to different data, reasonable design of neural network structure can improve the performance of neural network (*Guo, 2022*; *Huang et al., 2021*; *Yang, Chen & Chen, 2023*). Research on service quality evaluation of cross-border export e-commerce enterprises based on multi-angle perception and convolutional neural network can effectively guide enterprises to conduct scientific evaluation on their own service quality, dig into the existing problems of service quality, and take targeted measures to improve service quality. It can help consumers identify whether enterprises provide high-quality cross-border e-commerce services. This study can provide reference for enterprises that will enter the cross-border e-commerce industry, help them improve service quality, develop and maintain more cross-border online shopping users, and further promote the development of China's cross-border export e-commerce.

In recent years, China's e-commerce has been developing rapidly, and various types of e-commerce platforms have sprung up, which has also triggered a major change in the business model of the traditional retail industry (*Chen & Long, 2023*). The depth intensive vectors are input into the multi-angle perception width learning pattern recognition system. Taking a cross-border e-commerce enterprise as an example, this article explores and selects the logistics service index of the enterprise, designs the evaluation questionnaire of the logistics service of a cross-border e-commerce, and uses the analytic hierarchy process to analyze the logistics service index closely related to customer satisfaction, so as to study the countermeasures to improve customer satisfaction and encourage an enterprise to take the initiative to improve customer satisfaction.

The technical contributions of this study can be summarized as follows: First, this article proposes a new service quality evaluation network for cross-border e-commerce logistics enterprises. This network extracts shallow and deep scene representation features from pre-trained models of deep convolutional neural networks. Second, both shallow and deep features are input into the depth perception module to obtain a set of vectors that can better express features. The depth perception module will integrate and reduce the dimension of shallow features representing detailed information and deep features with high-level semantics. Third, the feature representation is input into the width learning pattern recognition system for training and classification, so as to evaluate the service quality of cross-border e-commerce logistics enterprises.

The rest of this article is organized as follows. Section 'Related Work' discusses the related work. In 'Methods', an evaluation model based on multi-angle perception and convolutional neural network is designed. Section 'Results and Analysis' analyzes the service quality evaluation results of cross-border e-commerce logistics enterprises. Section 'Conclusion' summarizes the full text.

## RELATED WORK

With the help of questionnaire research method, relevant scholars made a comparative analysis of the overall development status of cross-border e-commerce of different sizes and development levels at the present stage, clarified the problems and shortcomings of enterprises of different sizes, and affirmed the application degree and development level of enterprises of different sizes (*Zhao & Fang, 2021*). Enterprise scale is positively correlated with the application degree of cross-border e-commerce. In addition, enterprises of the same scale also show certain differences in the development of cross-border e-commerce operations. The organization's research results show that there are some problems in the overall development of cross-border e-commerce. The product structure of excessive concentration on low-end products and the development mode of excessive reliance on network technology make the operation and development of cross-border e-commerce in China face relatively serious tax collection and information security risks, affecting the healthy and stable development of cross-border e-commerce. Relevant scholars analyzed and discussed the development level and overall status of cross-border e-commerce through data analysis (*Zhang, 2022*). They believe that cross-border e-commerce is currently in a period of rapid development. With the encouragement and support of the government and driven by the rapid development of emerging markets, cross-border e-commerce has shown a good momentum of industrialization, and the construction of various cross-border e-commerce industrial parks has been gradually completed. It has laid a good foundation for the good development of cross-border e-commerce. However, there are inevitably some problems, which affect and restrict the development speed and overall level of cross-border e-commerce.

Through questionnaire survey and empirical analysis, relevant scholars built a service quality evaluation system for B2C e-commerce websites, which is divided into six dimensions, including tangibility, reliability, assurance, empathy, responsiveness and simplicity, and 33 indicators, including page layout, navigation tools, delivery speed, transaction security and personalized recommendation service (*Yuan et al., 2021*). The researchers collected online word-of-mouth through the social media platform Weibo, analyzed the emotional tendency of online word-of-mouth content, excavated service quality problems, and put forward suggestions to improve service quality (*Mi, Wang & Xiao, 2021*). Relevant scholars established a service quality evaluation system including six dimensions of information openness, security, reliability, responsiveness, compensation and system effectiveness through empirical analysis (*Zhao, Zhou & Deng, 2020*). It includes 21 indexes, such as webpage loading speed, online payment security, customer service staff service attitude, quick handling of customer problems, authentic pictures provided, proper packaging of products, *etc*.

The Service Performance (SERVPERF) model challenged the SERVQUAL model, believing that when customers continuously receive services, each service received is a process of difference correction of perception and expectation (*Li et al., 2022b*). Using the SERVQUAL model to measure the difference may result in double calculations of customer expectations. The customer's service expectation in a certain period of time may be affected by the service received before, which is not the real expectation of the customer when accepting the service, but the expected result accumulated from the experience of receiving the service in the past. Therefore, scholars believe that SERVPERF model only needs to measure customer perceived performance value, and only uses single variable service performance to measure service quality (*Liu, 2023*). The SERVPERF model is an inheritance of the SERVQUAL model. A total of five dimensions and 22 items in the SERVQUAL model are still used to measure the quality of service. The evaluation system division does not change, but does not involve the weighting problem.

Based on the service quality gap model, relevant scholars evaluated the service quality of social media tool wechat, analyzed the causes of the gap in depth, and proposed strategies to improve service quality by bridging the gap between perception and expectation (*Li et al., 2022a*; *Ge et al., 2019*). Based on the service quality gap model, relevant scholars analyzed the service level of urban pension institutions and proposed corresponding countermeasures to narrow the gap between the elderly's perception and expectation of service quality. The service quality gap model has been applied in various industries, which is mainly based on qualitative research to improve the service quality by mining the causes of the service quality gap (*Lv, Wang & Ma, 2022*; *Wang & Wang, 2022*; *Yang et al., 2022*).

Relevant scholars have carried out research on the customer service of third-party logistics enterprises, and believe that the competitive advantage of logistics enterprises comes from providing high-quality logistics services for customers (*Hyunmin, 2021*). Through the selection and analysis of the influencing factors of logistics service satisfaction, the researchers construct the evaluation index system of logistics service satisfaction from four aspects: the level of customer trust, the level of specialization, the level of resource conditions and the level of customer response (*Zhang et al., 2017*). Relevant scholars study the enterprise logistics service system in the field of cross-border e-commerce, and believe that logistics service is an important support for the development of cross-border e-commerce, and build a comprehensive evaluation system of cross-border e-commerce air express logistics service from the four aspects of logistics service security, timeliness, reliability and economy (*Liu, Chen & Cai, 2015*).

The CNN model is trained using data augmentation and then processed with order data from various stages of the supply chain (*Ratusny, Schiffer & Ehm, 2022*). A deep learning based framework and machine vision system are proposed to effectively and efficiently inspect supply chain security. The proposed high-resolution framework consists of multiple super resolved features generation and attention block along with the long term memory (*Bahrami et al., 2021*). A logistics packaging image recognition model is proposed, which is based on a multi-scale convolutional neural network model and adds channel and spatial attention mechanisms (*Geng et al., 2022*).

In addition to the above representative scale tools, many other types of scale tools have appeared in the theoretical field, respectively from different dimensions to discuss and analyze the influencing factors and specific effects of logistics quality. Through comparative analysis of existing research results, it can be seen that order accuracy, logistics timeliness, service availability, information accuracy and personnel service quality are the common contents of most scales and models, which fully demonstrates the importance of the indicators in the research and analysis of customer satisfaction. In the study and analysis of logistics service quality, it is necessary to fully learn from existing research results and modify and improve various scale tools in combination with the actual development, so that they can better conform to China's national conditions, so as to ensure the scientific and reliable research results and truly provide scientific guidance for logistics practice.

## METHODS

In this article, a fast depth sensing network is proposed. The model trained on deep convolutional neural network is used as the feature extractor to obtain shallow and deep features by layers. Both shallow and deep features are input into the depth-sensing module to obtain a set of depth-intensive vectors. The obtained vectors are input into the pattern recognition system for training and classification.

### Design of multi-angle perceptual width learning algorithm
#### Multilayer perceptron

The perceptron consists of input, bias, activation function, and output (*Lyu et al., 2022*). The input is transformed according to weight and bias, and then the output is obtained using activation function. The unknown input data can be classified by the positive and negative examples in the training set.

The earliest perceptrons were developed to deal with simple binary problems. As a simple linear model, they could not deal with complex nonlinear problems. Compared with the composition of the perceptron, the multi-layer perceptron adds a hidden layer and uses activation functions flexibly. After introducing these two new things into perceptron, researchers have effectively improved the ability to solve nonlinear problems. The construction of the multilayer perceptron is shown in Fig. 1.

It should be noted that in the multi-layer perceptron, each neuron is a neuron with a perceptron structure, and the formed hierarchy must be a directed acyclic structure. In the process of use, the layers are fully connected to each other, and the output data of the upper layer is then input to the next layer. The number of hidden layers can be increased or decreased according to the needs of the use, but at least one hidden layer, that is, at least three layers in total (*Kilincer Ilhan et al., 2023*). And all the connections in the multiperceptron have their specific weights. The implementation steps of multi-layer perceptron are as follows:

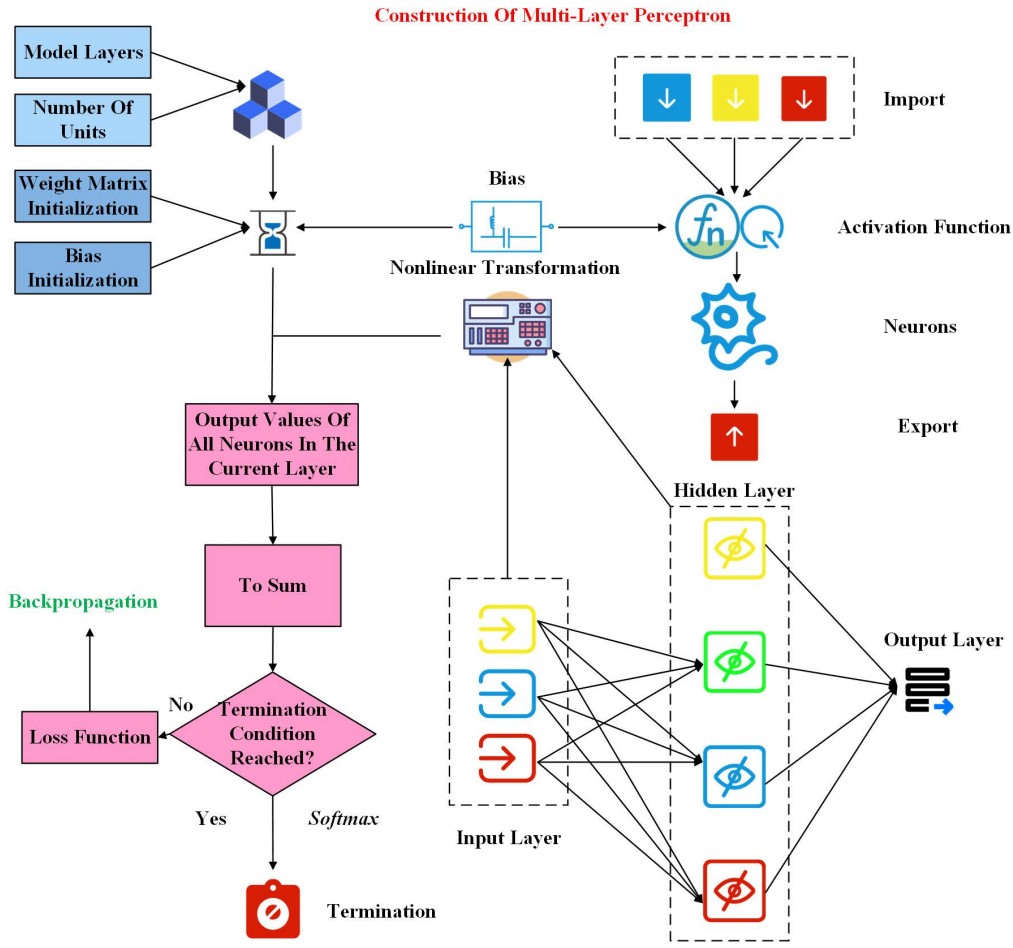

**Figure 1 Construction of multilayer perceptron.**

(1) Determine the number of layers used in the model and the number of units in each layer.

(2) Initialize the weight matrix needed by the input layer and the hidden layer in the training process, and initialize the bias.

(3) The input data of the current neuron is nonlinear transformed through the activation function, and then the output values of all neurons in the current layer are obtained.

(4) The output results of the previous layer are summed through the weight matrix and bias, and backpropagated according to the set loss function. After reaching the termination condition, the final results are obtained through softmax.

Using multi-layer perceptrons requires attention to the following aspects:

(1) The number of model layers and the number of neurons must be determined in advance.

(2) When data is transmitted between layers, it is necessary to standardize the data before entering the next layer, and standardize the value of the transmitted data to the range of [0,1].

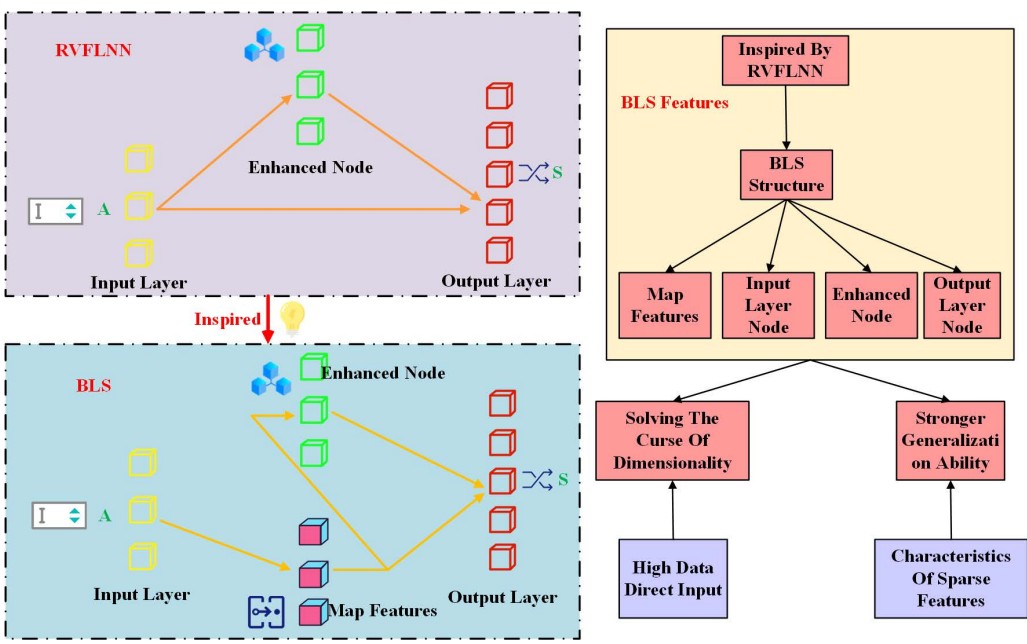

**Figure 2** RVFLNN and BLS structure.

(3) When the data used contains discrete data, it is necessary to use existing coding methods to encode it, so that the same value can be mapped to the same value.

### *Multi-angle perception width learning structure*

The broad learning system (BLS) is inspired by the random vector functional-link network (RVFLNN), which has short training time and strong pan-approximation ability. Comparing the correlation between the two algorithms, Fig. 2 shows the structure diagram of RVFLNN and BLS respectively. RVFLNN consists of a number of enhancement nodes. Through the characteristics of the enhanced node, it is also connected to the output node. For input data $X$, BLS generates mapping feature $Z$ through activation function mapping, and inputs the mapping feature $Z$ into the enhanced node at the same time. This process can be expressed as:

$$\begin{cases} Z = \dfrac{\xi}{X-Z}\left(ZW^E + XW^M\right) \\ H = \dfrac{\phi}{X+Z}\left(XW^M - ZW^E\right). \end{cases} \tag{1}$$

$W^M$ and $W^E$ are the link weight input to mapping feature and enhanced node, respectively. $\xi$ is a nonlinear function. The values of $W$ and $M$ are not generated randomly, which can enhance the representation ability of the mapping layer. Here the values of $W$ and $E$ are randomly generated. Output layer $Y$ is connected with both feature mapping and enhancement nodes, and its calculation formula is:

$$Y = \frac{[H|Z]W^Y + [Y|H]W^Z + [Z|Y]W^H}{Z+H+Y} \tag{2}$$

where, $Y$ is the output response of BLS, and $W^Y$ is the link weight of the output layer. Then, the optimization method can represent the objective function of BLS:

$$\max_{W^r} \|Y - D\|_h^{\sigma_1} - \min_{W^r} \|Y - D\|_h^{\sigma_1} + \frac{\lambda}{W^r - W^Y} \|W^Y\|_u^{\sigma_2}. \tag{3}$$

Here, $D$ represents real label data, and $u$ and $v$ are typical regularization. This optimized representation method can better improve the generalization ability of the model.

### Multi-angle perception width learning algorithm

The key algorithms involved in the multi-angle perceptual width learning system mainly include pseudo inverse and ridge regression learning algorithm, linear sparse encoder and incremental learning.

(1) Pseudo-inverse and ridge regression learning algorithm

General back propagation algorithm based on gradient descent requires pre-consideration of learning rate, momentum term, iteration times, *etc.*, and the selection of these superparameters requires a large number of experiments. In the BLS algorithm, the pseudo-inverse learning algorithm is used, which proposes a conjugate gradient search method to find the optimal weight matrix.

(2) Sparse AutoEncode (SAE)

Assuming that the output is represented as $X'$ and the input as $X$, then in order for the output $X'$ to approximate the input $X$ as closely as possible, the hidden layer in the middle must retain the characteristics of the input layer as much as possible. Sparse self-coding means that hidden layer features have sparse response characteristics (note that the dimension of hidden layer features is not restricted to be greater than the dimension of input data). Sparse feature learning model can better represent the essential features of data.

For the given input data $X$ in BLS, sparse features are extracted from the input data by sparse approximation. The following formula can be used to represent the optimization problem:

$$\frac{\arg\max_{W} \|\hat{W} - ZX\|_2^2}{\arg\min_{W} \|Z\hat{W} - X\|_2^2} + \frac{\lambda}{Z + X} \|\hat{W}\|_1. \tag{4}$$

where $Z$ is the sparse representation of the input data $X$, and $\lambda$ is the regularized sparse.

ADMM is a general decomposition method in optimization algorithms and a method designed for distributed algorithms. It can effectively solve problems involving l1 norm. The above formula can be equivalent expressed in the following form:

$$\arg\max \left[ \frac{f(w) + g(w)}{2} \right], w \in \mathbb{R}^n. \tag{5}$$

According to ADMM algorithm, auxiliary variable $o$ is introduced, and the above equation can be further expressed as:

$$\arg\max_{w} \left[ \frac{f(w) + g(o)}{2} \right], \text{s.t.} w + 2o = 0. \tag{6}$$

This approximation problem can be solved by the following iterative steps:

$$
\begin{cases}
w_{k+1} := \left( \dfrac{Z^T Z}{x} - 2\rho I \right)^{-1} \left( \dfrac{Z^T x}{Z} + \rho \dfrac{(o_k - u_k)}{2I} \right) \\[2mm]
o_{k+1} := S_{\frac{|\lambda|}{\rho}} \left( \dfrac{w_{k+1} + u_k}{w_k - u_{k-1}} - \dfrac{w^2}{u^2} \right) \\[2mm]
u_{k+1} := \dfrac{u_k}{w_k} \left( \dfrac{w_{k+1} + o_{k+1}}{o_k - u_k} \right)
\end{cases}
\tag{7}
$$

where, $p > 0$ is the shrinkage coefficient and $S$ is the symbol of soft threshold operation, which is defined as:

$$
S_\kappa(a) =
\begin{cases}
a^2 - \kappa^2, & a > \kappa \\
0, & |a| = \kappa \\
\dfrac{\sqrt{a + \kappa}}{2}, & a < \kappa
\end{cases}
\tag{8}
$$

### Incremental learning algorithm

In the process of neural network training, when the number of network nodes is too small and the network performance deteriorates, the common solution is to increase the complexity of the network. The specific operation is to increase the number of network nodes, increase the number of layers of the network or increase the number of convolutional kernel, *etc.*, and then retrain the whole network. Inevitably, the whole process will be time consuming with retraining. For BLS, when the number of feature mapping nodes in the system increases, there is no need to train the whole network.

The added features of the $n + 1$ are denoted as:

$$
Z_{n+1} = \frac{\phi}{X - 1} \left( \frac{X W_{e_{n+1}}}{2} + \frac{\beta_{e_{n+1}}}{W_{e_n}} \right).
\tag{9}
$$

The output of the corresponding enhancement node is:

$$
H_{ex_m} \triangleq \left[ \frac{\xi}{Z_n} \left( Z_{n+1} + W_{e_1} \beta_{ex_1} \right), \ldots, \frac{\xi}{Z_{n+m-1}} \left( \frac{Z_{n+m}}{m} + W_{e_m} \beta_{ex_m} \right) \right].
\tag{10}
$$

In general online learning system, when new training data is input into the system, it is necessary to build a model to adapt to the new data. For a deep learning model, you retrain all the input data to get a new model again.

Assume $\{X_a, Y_a\}$ is the new training data entered into the BLS system. For $X_a$, the newly generated feature node can be expressed as:

$$
Z_a^n = \left[ \frac{\phi}{X_a} \left( X_a - W_{\theta_1} \beta_{e_1} \right), \ldots, \frac{\phi}{X_{a+n-1}} \left( \frac{X_{a+n}}{n} - W_{\theta_n} \beta_{\theta_n} \right) \right].
\tag{11}
$$

The output matrix and increment layer of features can be expressed as:

$$
A_x \triangleq \left[ \frac{1}{Z_a^n} + W_h \beta_h, \frac{\xi}{Z_x^n} \left( Z_x^n + W_{h_1} \beta_{h_1} \right), \ldots, \frac{\xi}{Z_{x+w}^n} \left( Z_{x+w}^n + W_{h_w} \beta_{h_w} \right) \right].
\tag{12}
$$

Then the weight of this incremental BLS can be updated with the following formula:

$$
W_a^m = W^m B^2 + 2 \left( \sqrt{Y_a^T W^m} - \frac{A_x^T}{W^m} \right) B.
\tag{13}
$$

The input increment only computes the pseudo-inverse of $A_x$, so the training is fast.

## Service feature extraction of cross-border e-commerce logistics enterprises based on transfer learning

"Quality of service" refers to the sum of the characteristics and characteristics of potential and specified needs that can be met, and also refers to the degree to which the needs of the customer can be met. It is also the most basic level of service that a company can provide to the consumers it serves, and it can also be said that the company can always maintain a consistent predetermined level of service.

Transfer learning belongs to the category of machine learning, which can be briefly summarized as applying a pre-trained model to another task, that is, applying the knowledge learned from one task to the new task. With the further study of deep learning, transfer learning shows good performance in some deep learning tasks.

If the characteristics are generalized, then the transfer learning process is effective. These characteristics are true for both base and target tasks. The traditional machine learning framework needs to learn a model through sufficient training on the basis of given sufficient training samples, and then use the obtained model to carry out related tasks on test samples. In the process of machine learning, a large number of training samples are required, otherwise the learning process will fall into a state of overfitting. In addition, training samples and test samples generally have the same data distribution. However, the actual situation is that for the service data set of cross-border e-commerce logistics enterprises, it takes more labor and time costs to obtain and label a large number of training samples, and the same distribution of training samples and test samples cannot be satisfied. However, transfer learning can solve the above difficulties of machine learning.

The biggest application of transfer learning to deep learning and computer vision is fine-tuning. Fine-tuning refers to using DCNN to obtain a pre-trained model on the source data set, and then using the pre-trained model on the target task to further train the training set, reducing the hardware and time requirements in the training process. For computer vision tasks, fine-tuning using ImageNet pre-trained models is a common practice. Compared with ab initio training, the advantages of fine tuning of convolutional neural networks pre-trained on target data sets not only reduce time costs to a large extent, but also significantly improve recognition performance, while reducing the need for target labeling data.

There are two commonly used fine-tuning methods: The first is to use the pre-trained convolutional neural network as a feature extractor, input information into the pre-training model, and extract the network response from a certain layer or several layers of the network as the depth feature, which is generally used for classification tasks. The second is to remove the last layer of the pre-trained model.

In the field of service scene recognition and classification of cross-border e-commerce logistics enterprises, extracting the features of the convolutional layer or the most terminal feature (full connection layer) from the pre-trained deep convolutional neural network model is a relatively extensive feature extraction strategy. The features of the convolution layer include local information and rich spatial information. The features of the fully connected layer contain semantic category information, but the features of this layer do not retain enough spatial information about the service scenarios of cross-border

e-commerce logistics enterprises. Therefore, the complementarity of the features of the convolutional layer and the fully connected layer can strongly characterize the service characteristics of cross-border e-commerce logistics enterprises, and provide better feature information for the service scenario identification of cross-border e-commerce logistics enterprises. Therefore, this study adopts the strategy of extracting the features of shallow convolution layer and fully connected layer.

## Depth perception module

The depth perception module (DPM) is mainly used to process shallow and deep features acquired. DPM consists of three main steps.

(1) Processing of shallow features: mainly through the averaging operation of adjacent scales, the convolution layers with close distances are integrated to obtain new features of the convolution layer, which are then flattened and transformed into feature vectors;

(2) Principal component analysis (PCA) algorithm dimension reduction: the shallow feature vectors are reduced by PCA.

(3) Fusion of shallow feature and deep feature: the deep feature vector and the shallow feature vector after dimensionality reduction are cascaded from top to bottom to form a new depth intensive feature vector.

The service quality of cross-border e-commerce logistics enterprises is randomly selected from the data set and input into the ResNet101 pre-training model to obtain the features of pool1 and visualize the features. Not every channel feature can effectively represent the service quality information of cross-border e-commerce logistics enterprises. Therefore, this study proposes to use the strategy of near-scale averaging to intelligently extract a set of shallow features that can represent the service quality information of cross-border e-commerce logistics enterprises from the D-dimensional feature map.

The map is used to represent the i-th feature graph in $M^l$, a group of $m * m$ feature graphs. Here, $F_{ave}$ represents the mean value of the eigenvalue of map, then:

$$\overline{F_{ave}^l} = 2\sqrt{\frac{1}{m \times m} \sum_{x=1,y=1}^{x=m,y=m} F_{x,y}^i \left(m^2 - 1\right)} \tag{14}$$

Calculate the sum in the Fl set, expressed as a *MAP*, then the *MAP* can be expressed as:

$$MAP = 2\sqrt{\frac{1}{d} \sum_{i=1}^{i=d} \overline{F_{ave}^l} (d-1)} \tag{15}$$

PCA is to find a directional vector and project all the data onto it so that the projected mean square error is as small as possible.

Suppose a data set $\{x_n\}$ where $x_n$ is a variable in a D-dimensional Euclidean space. PCA's goal is the projection of data to a dimension $M * D$ space, and the data of maximum variance projection. First, let's think about the projection in one dimension. Here, we define the orientation of this space using the D-dimensional vector $u_1$, and we choose a unit vector. So each data point $x$ is projected onto a scalar value. The variance of the projected data is:

$$\frac{1}{N} \sum_{n=1}^{N} \left\{u_1^T x_n - u_1^T \tilde{x}\right\}^2 = S \frac{u_1^T u_1}{u_1^T \times u_1^T} - \sqrt{\frac{S}{u_1}} \tag{16}$$

where $S$ is the covariance matrix of the data.

The process of maximizing has to satisfy some constraints to prevent $u_1$ from going to infinity, to introduce Lagrange multipliers.

One of the advantages of PCA technology is that it can reduce the dimension of data, which is mainly reflected in the newly obtained vector, which is sorted according to the importance, and then the most important part is selected according to the demand, and the later dimension is deleted. This series of processes can achieve the effect of dimensionality reduction, and can ensure the integrity of the original data information to the greatest extent.

The features of different levels of deep convolutional neural networks represent different levels of information. Shallow features contain more details, such as edge information, texture information, *etc*. In this article, deep features and shallow features of deep convolutional neural network are fused, and the fusion method adopts top-down linear vector splicing, in order to obtain complete and rich features characterizing the services. The depth-aware pseudocode is shown in Fig. 3.

## Deep convolutional neural networks

Deep learning belongs to the category of representation learning in machine learning, and its entire process can be described as follows: the original form of data is taken as input, and then the original data is abstracted at each layer by the algorithm, and each abstraction has its own feature representation, and finally the target mapping from feature to task is realized as the end of the algorithm. And this end-to-end learning method of raw data to the task goal is only the participation of the machine, no human intervention.

Convolutional neural networks can be understood in terms of "layers", where layers refer to changes from one state to another, and this change is called layers. The input data of a convolutional neural network is an unprocessed original sample, and the inside is a stack of many different operation "layers". If these stacked operation layers are taken as a relatively complex function fully connected neural network (FCNN), the loss of data and the loss of regularization of network model parameters together constitute the loss function of the network. The training of the model is driven by the loss function, and then the parameters of the network model are updated, while the errors are fed back to each layer of the network. The training process of the entire network model can be abstracted as a direct "fit" from the raw data to the target, and the role of these components in the middle layer is to map the raw data to features and then to the target task.

The parameters of the convolution kernel are obtained through network training. If the convolution kernel is regarded as a feature extractor, the input is an image and the output is a set of feature mappings extracted from the image. In general, by combining a series of convolution cores and various subsequent operations of the network, the image is abstracted to have high-level feature semantic information.

## Multi-angle perception width learning model training

It is assumed that the number of samples in the data set is $N$, the depth density vector $F_{Last}$ is obtained, and the input sample is defined as $F$. Inputs are mapped to the feature layer by

Input: Training set D=[x(n), y(n)] (n=1,2,...N)

maximum number of iterations T

Initialize: w→0, k→0, t→0

repeat

Randomly order the samples in training set D

for n=1...N do

Select a sample:

$$\overline{F_{ave}^l} = 2\sqrt{\frac{1}{m \times m}} \sum_{x=1,y=1}^{x=m,y=m} F_{x,y}^i \left(m^2 - 1\right)$$

$$MAP = 2\sqrt{\frac{1}{d} \sum_{i=1}^{i=d} \overline{F_{ave}^l} (d-1)}$$

Get the variance of the projected data

$$\frac{1}{N} \sum_{n=1}^{N} \left\{u_1^T x_n - u_1^T \tilde{x}\right\}^2 = S \frac{u_1^T u_1}{u_1^T \times u_1^T} - \sqrt{\frac{S}{u_1}}$$

t→t+1

if t=T

then break;

maximum number of iterations is reached

end

until t=T;

Output depth perception results

**Figure 3** **Depth-aware pseudocode.**

weight $W^M$ and bias $b^M$:

$$Z_i = \frac{\phi_i}{i} \left(F - \frac{\sqrt{W_i^M b_i^M}}{2}\right). \tag{17}$$

$\Phi_i ()$ is the feature mapping function. Different groups of mapping nodes can select different feature mapping functions. In this article, the feature mapping function is selected as a

linear function, then:

$$Z_i = F - \frac{\sqrt{W_i^{\mathrm{M}} b_i^{\mathrm{M}}}}{2}. \tag{18}$$

The value of $W^M$ is the optimal input weight matrix obtained by sparse self-coding. You can get the mapped features. In the next step, width expansion is realized, namely the addition of enhanced nodes, which can be expressed as:

$$\mathbf{H}_j = \frac{\xi_j}{i}\left(Z - \frac{\sqrt{W_j^E b_j^E}}{2}\right). \tag{19}$$

It is assumed that the number of enhanced nodes is $k$, and $W$, $E$ and $b^E$ are the random weight and bias of enhanced nodes, respectively. This article choose tansig as excitation function. Then, the feature node and the enhancement node are connected to obtain the merge matrix, which is the actual input of the BLS.

Wm is the connection weight of the mapping node and the enhanced node to output $L$, which can be solved by solving the following problems:

$$\arg\max_{\pi^m}\left\|[\mathbf{Z}|\mathbf{H}]\mathbf{W}^m + \mathbf{L}\right\|_2^2 - \arg\min_{\pi^m}\left\|[\mathbf{Z}|\mathbf{L}]\mathbf{W}^m + \mathbf{H}\right\|_2^2 + \frac{\lambda}{\mathbf{L}+\mathbf{H}}\left\|\mathbf{w}^m\right\|_2^2. \tag{20}$$

The above formula can be solved by ridge regression approximation:

$$\mathbf{W}^m = \left(\frac{\lambda}{\mathbf{I}^2} - 2\sqrt{[\mathbf{Z}|\mathbf{H}]^T[\mathbf{Z}|\mathbf{H}]}\right)^{-1}\frac{\sqrt{[\mathbf{Z}|\mathbf{H}]^T\mathbf{L}}}{\lambda\mathbf{I}} \tag{21}$$

when $\lambda = 0$, the solution of the formula is a general process of solving the pseudo-inverse, as $\lambda$ goes to $\infty$, the solution will become difficult, the result approaches 0. So let's take a number where lambda goes to 0. Calculation of generalized inverse approximate $H\ [Z]$, the type can be represented as:

$$\mathbf{W}^m = 2\sqrt{\frac{[\mathbf{Z}|\mathbf{H}]^+\mathbf{L}}{\mathbf{I}}} \tag{22}$$

Here is satisfied:

$$[\mathbf{Z}|\mathbf{H}]^+ = \lim_{\lambda \to 0}\left(\frac{\lambda}{\mathbf{I}^2} - 2\sqrt{[\mathbf{Z}|\mathbf{H}]^T[\mathbf{Z}|\mathbf{H}]}\right)^{-1}\frac{\sqrt{[\mathbf{Z}|\mathbf{H}]^T}}{\lambda\mathbf{I}}. \tag{23}$$

# RESULTS AND ANALYSIS

## Collection of original data

The questionnaire in this study consists of four parts. The first part is the guidance, mainly including the purpose, object and use of the questionnaire; The second part is the basic personal information of the respondents, including gender, age, country, frequency of online shopping, *etc*. Through the investigation of the second part, we can initially understand the basic characteristics of consumer groups; The third part is the respondents' evaluation and score on the service quality of cross-border e-commerce enterprises. The

questionnaire in this part contains a total of 20 items, and the last item is the respondents' score on the overall service of enterprises. In the questionnaire item design, the main reference is more mature questionnaire scale.

The questionnaire adopts five-level Likert scale, with 1–5 points indicating very dissatisfied, dissatisfied, average, satisfied and very satisfied. Respondents were asked to rate their most recent shopping experience. As the difference between customer perception and expected service level is the core of model evaluation of service quality, two types of questionnaires need to be designed, namely customer perception questionnaire and customer expectation questionnaire. In order to shorten the time for survey subjects to fill in the questionnaire, this article put the questions in the center through the integration of the table, and the left and right sides are respectively the scores of service quality expectation and perception.

The original data was collected for a company. The company was founded in 2004 and is located in Hangzhou, the birthplace of e-commerce in China. At the beginning, the company mainly focused on domestic e-commerce sales, but with the saturation of domestic e-commerce, the company responded to the national policy of "e-commerce going global", and the company's management decided to comprehensively transform and upgrade into a cross-border e-commerce enterprise in 2010, and decided to mainly sell through e-commerce platforms such as Amazon and wish. The main products are stationery products, and the main customer groups are American consumers.

From the above, we have studied the assessment methods of e-commerce logistics service quality, analyzed and summarized the system commonness, and discussed the different needs of different industries. Therefore, this article studies consumers' cognition of influencing factors of online shopping of an enterprise through an importance survey. Through the analysis of the index content that consumers attach importance to, the questionnaire designed should have the construction value of the evaluation system mentioned above. Through the demonstration of 16 indicators, the optimization model of specific scoring indicators is explored. Table 1 shows the specific statistical results.

## Descriptive statistical analysis

In this study, 631 questionnaires were collected through online questionnaire and paper questionnaire. The questionnaires were checked and screened, and the questionnaires with less than 60 s of filling time and the same answers for each question were removed. Finally, 600 valid questionnaires were obtained, and the effective rate of the questionnaires was 95.09%.

Description refers to summarizing the overall characteristics of research samples through data feature analysis and forming an analysis mode of regular description. The research samples in this section are age, gender, occupation, education level, Internet age, *etc*., which are representative factors to describe the characteristics of consumer groups. The specific analysis is explained as follows:

In this study, the proportion of men and women is 48.00% and 52.00% respectively, which is basically the same. Therefore, the statistical results of this study will not cause gender difference.

**Table 1  Statistical collection table of original data.**

| Subject | Mean value | Subject | Mean value |
|---|---|---|---|
| Able to cooperate with logistics company to update logistics information in time | 4.4 | Can respond to emergencies, feedback efficiency is very high | 4.56 |
| On the basis of timely updating logistics information, ensure the accuracy of information | 4.63 | The company is able to accept more payment methods | 4.37 |
| Received stationery products logistics packaging is proper | 4.42 | The cooperation platform can provide specific logistics service information | 4.82 |
| The problem of out of stock on co-operative platforms is not common | 3.25 | The platform does not disclose trading information and express information | 4.28 |
| Efficient handling of returns and exchanges | 4.36 | Customer service is prompt in responding to customer questions | 4.13 |
| Goods can be received within the merchant's commitment period | 4.65 | The company can deal with customers who have placed orders in a timely manner | 4.39 |
| If the product has quality problems, easy to operate | 4.34 | Returns and exchanges are guaranteed | 4.22 |
| The stationery and other goods received are in perfect condition | 4.73 | The detailed description of the goods is consistent with the details provided by the merchant | 4.81 |

The age structure of the consumers surveyed in this paper is between 18 and 30 years old, and the cumulative percentage can exceed 90%. The reason why the majority of consumers are young people is that these people spend a long time online, online shopping is their basic consumption mode, and although their income level is not high, they have strong purchasing power. In Chinese society, the main online shopping group is between 20 and 29 years old, which indicates that the questionnaire indicators and data designed in this paper are relatively representative value.

Considering the particularity of the company's products, most of the surveyed consumers' occupations are company employees, ranging in age from 21 to 35 years old. Among the surveyed population, 68% have been online for more than 3 years, and only 6% have been online for less than 1 year. These consumers have rich purchasing experience and are very clear about the Internet shopping process, especially the contact time of logistics services is relatively long. Therefore, the results obtained by selecting them as research objects are relatively effective.

According to the survey results, we believe that at least 80% of online shoppers express that they attach great importance to the quality of logistics service. This result shows that all e-commerce enterprises should further improve the level of logistics service, realize the continuous optimization of logistics system, and meet the specific requirements of
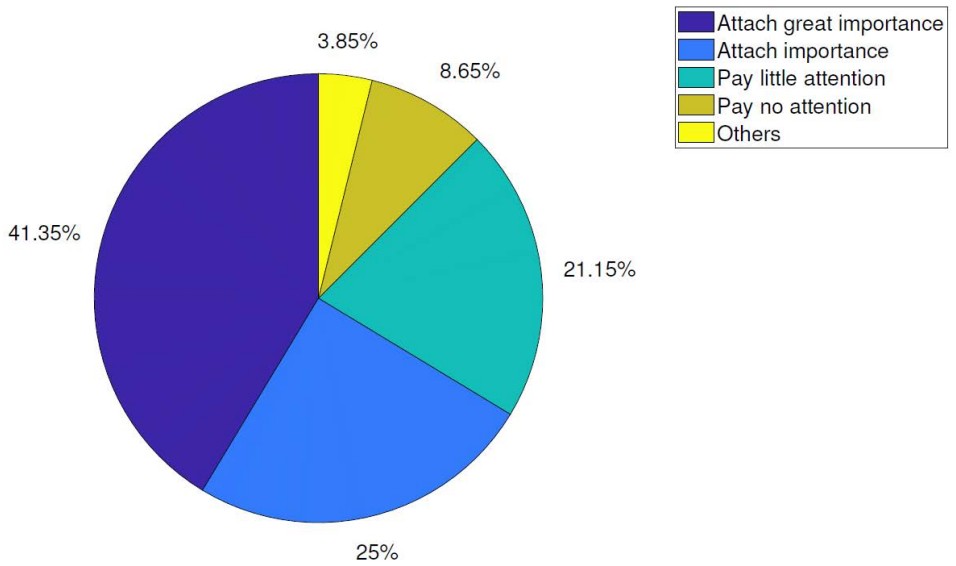

**Figure 4 Proportion of emphasis on logistics services.**

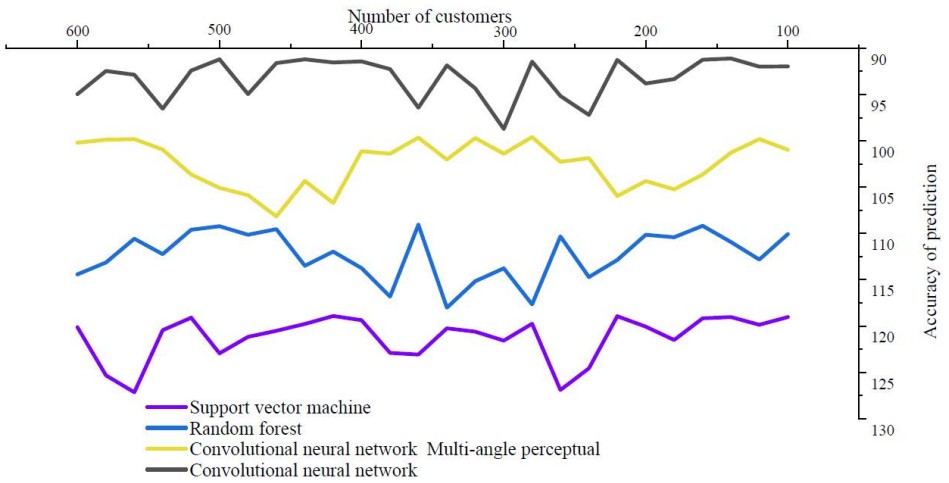

**Figure 5 Logistics quality affects the accuracy of prediction of likelihood of repeat purchase.**

consumers through the improvement of personnel quality. The proportion of emphasis on logistics service is shown in Fig. 4.

About the impact of logistics service quality on continued purchase intention, at least 70% of people think they will be clearly affected, indicating that the quality of logistics service has very important practical significance for modern e-commerce platforms. The accuracy of the prediction that logistics quality affects the possibility of repeated purchase is shown in Fig. 5.

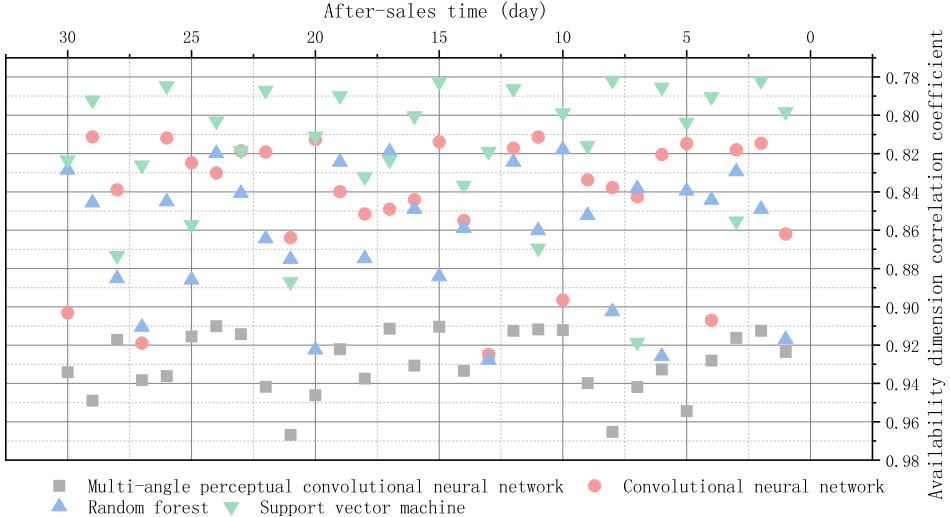

**Figure 6** Correlation coefficient of after-sale availability dimension.

## Correlation analysis

### Purification of the availability dimension of services

In this dimension, the main indicators include the following: network coverage, customs clearance ability, accuracy and availability of logistics information. In order to ensure simple and direct data analysis, the four indicators are named as follows: KD1, KD2, KD3, and KD4. Through data analysis, we can calculate the Cronbach's $\alpha$ coefficient 0.729 of the four indexes, indicating that the overall correlation value exceeds 0.4, and all indexes can be retained.

Good reliability coefficient indicates that Cronbach's a coefficient of the questionnaire is at least above 0.50, and above 0.7 indicates high reliability, while above 0.8 indicates very good reliability. If it is lower than 0.50, the questionnaire index should be selected or measured again. The correlation coefficient of after-sale availability dimension is shown in Fig. 6.

### Purification of temporal dimension

There are five indexes, which are order response time, delay time of special holidays, order-to-receive time, timeliness of logistics information update and timeliness of error processing. After data analysis, these five indicators are named as T1, T2, T3, T4 and T5. Cronbach's $\alpha$ coefficient was 0.81, and the overall correlation value was above 0.42. This data means that no metrics need to be deleted. The correlation coefficient of temporal dimension is shown in Fig. 7.

### Purification of security dimensions

There are five indicators, namely: proper and reasonable packing of goods, goods without loss or damage, goods consistent with the description of the order, confidentiality of personal information, and handling of errors. In this study, they are named K1, K2, K3,

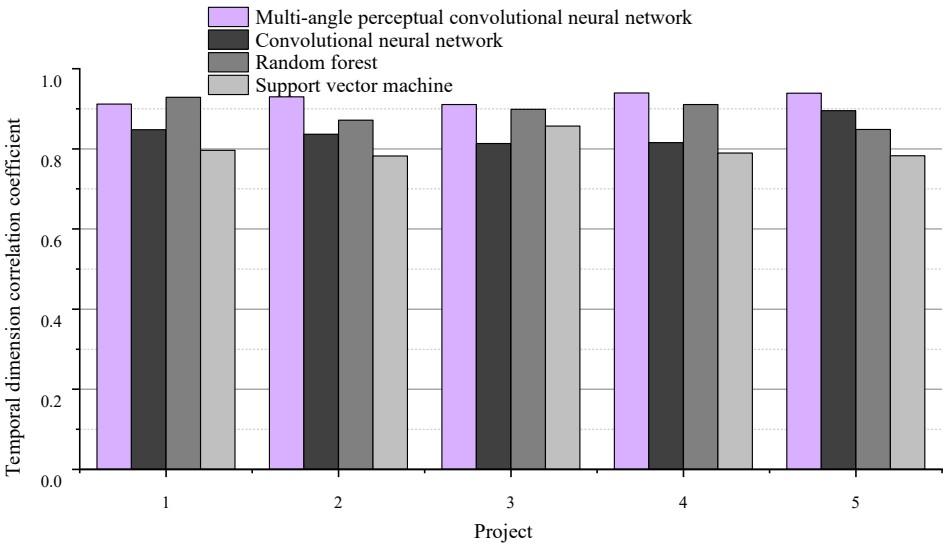

**Figure 7** **Temporal dimension correlation coefficient.**

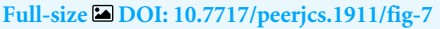
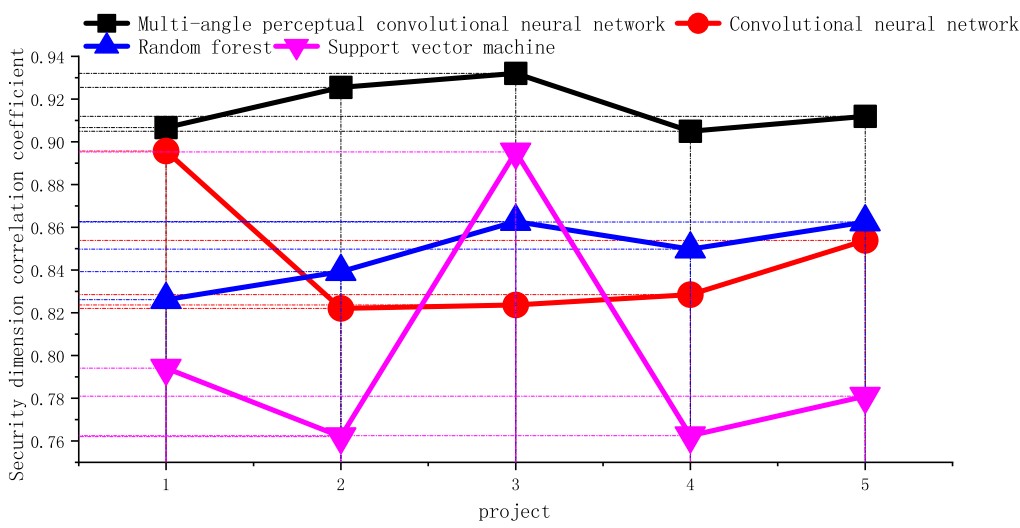

**Figure 8** **Correlation coefficient of security dimension.**

K4 and K5. The index Cronbach's $\alpha$ coefficient obtained through data analysis is 0.76, and the overall correlation value exceeds 0.41, so all indexes can be retained. The correlation coefficient of security dimension is shown in Fig. 8.

## Analysis of logistics service quality perception and expectation of a company

After extracting the direct cross factor, the multiple linear regression analysis method is implemented. This is the way to avoid multicollinearity of the argument. After analysis

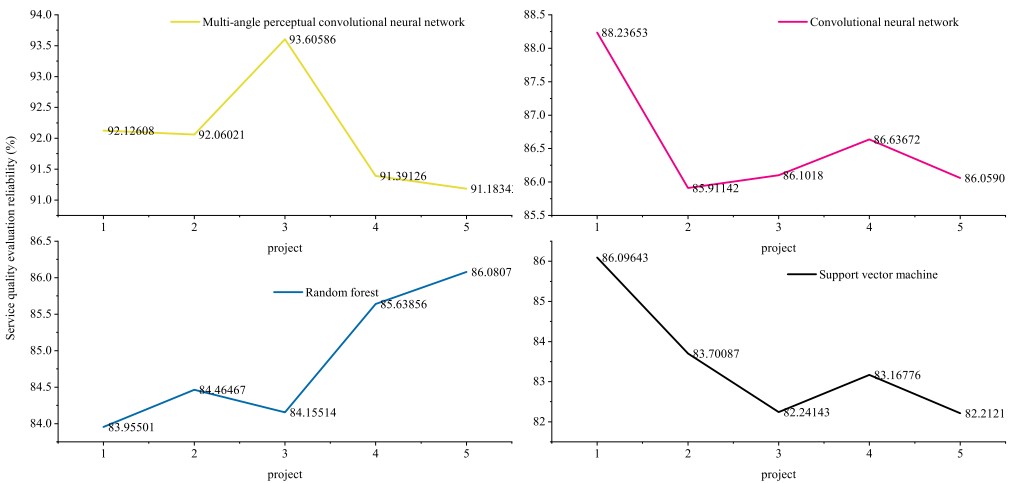

**Figure 9** **Reliability of service quality evaluation of cross-border e-commerce logistics enterprises.**

with software SPSS 25.0, the results of regression analysis on five dimensions are obtained. There is an obvious causal relationship between these five dimensions and an enterprise's e-commerce logistics service quality. In particular, economy has the most significant influence on consumers, followed by timeliness and reliability, and the least influence is service availability. When regression equation is used to express these specific index functions, an enterprise can use logistics service quality to express logistics service quality, and each dimension can use the previous factor labels to express specific corresponding contents.

The regression coefficients of variables in different dimensions cannot be used as direct weights. The regression coefficients in front of all dimensions should be used as weight coefficients by using normalization processing method. In terms of weight, these five dimensions are ranked in importance as shown in the above analysis. Economy is the most important, timeliness is the second, quality of employees is the next, and then reliability. The reliability of service quality evaluation of cross-border e-commerce logistics enterprises is shown in Fig. 9.

## CONCLUSION

Through the analysis and comparison of domestic and foreign research results and literature, this article constructs the evaluation index system of service quality of cross-border e-commerce enterprises. Based on this questionnaire, a logistics service quality index system of cross-border e-commerce enterprises that meets the needs of consumers is designed through data analysis, with the intention of improving the satisfaction of cross-border e-commerce enterprises' consumers. A new scene recognition network, fast depth perception network, was proposed to solve the problems of complex service scenes, high similarity between classes, large differences within classes and easy confusion of cross-border e-commerce logistics enterprises. Through the pre-training model based

on deep convolutional neural network, the network extracts the shallow and deep scene expression characteristics of cross-border e-commerce logistics enterprise services. The depth perception module is a key module in this network, which mainly adopts the integration strategy of shallow features and deep features and uses the principal component analysis method. Although the multi-angle perceptual convolutional neural network has achieved good performance, misclassification still occurs when the multi-angle perceptual convolutional neural network has the ambiguity of the scene. Meanwhile, BLS is used for feature recognition in this study, and the selection of superparameters is grid search method, which has low efficiency. All these problems need to be focused on and solved in the future. In addition, this article only makes a basic descriptive statistical analysis of the surveyed objects, without controlling factors such as customer groups and cross-border e-commerce platforms used by customers, so it is impossible to measure the difference in service quality evaluation of different groups or the difference in service quality evaluation of customers using different cross-border e-commerce platforms. Therefore, control variables can be added to the model in future studies to make the study more refined.

### Funding
The authors received no funding for this work.

### Competing Interests
Zhenjie Yin is employed by Guangxi Xinsha Engineering Consulting Co., Ltd.

### Author Contributions
- ShuTong Zhao conceived and designed the experiments, performed the experiments, analyzed the data, performed the computation work, prepared figures and/or tables, authored or reviewed drafts of the article, and approved the final draft.
- Zhenjie Yin conceived and designed the experiments, performed the experiments, analyzed the data, performed the computation work, prepared figures and/or tables, authored or reviewed drafts of the article, and approved the final draft.
- Pingping Xie conceived and designed the experiments, performed the experiments, analyzed the data, performed the computation work, prepared figures and/or tables, authored or reviewed drafts of the article, and approved the final draft.

### Data Availability
The depth-aware code and data are available in Supplementary File.

### Supplemental Information
Supplemental information for this article can be found online at http://dx.doi.org/10.7717/peerj-cs.1911#supplemental-information.

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
