# Peer review of "Multi-angle perception and convolutional neural network for service quality evaluation of cross-border e-commerce logistics enterprise"

_PeerJ Computer Science, doi:10.7717/peerj-cs.1911_

## Round 0.1 · original submission · Major Revisions

As per comments from 4 reviewers, this paper can get a major revision for further consideration.

Reviewer 1 ·

Basic reporting

Title: Multi-angle perception and convolutional neural network for service quality evaluation of cross-border e-commerce logistics enterprises
• Convolutional Neural Network was only mentioned in bits and pieces in section 3. There is no dedicated subsection for discussion on CNN.
• The title states “enterprises” but data was collected only from one single company.

Abstract
• Several statements in the abstract were not found on the paper:
o “Training and testing” were mentioned in the abstract but was not explained how it was performed in this research using CNN.
o “Excellent classification performance is obtained”. However, it was not mentioned in the paper what form of classification and how it was tested.
o “Traditional cross-border e-commerce logistics service scenario classification algorithm”. There was no mention of what is the traditional algorithms.

Related works
• This section has several limitations as it did not discuss the major topics related to the title.
• First, no discussion was made on the existing works related to Multi-angle perception and convolutional neural networks. The authors only include multi-angle perception in section 3 methodology with little (almost to none) references from the existing works. Convolutional Neural Network was only mentioned in bits and pieces in section 3. There is no dedicated subsection for discussion on CNN.
• Some of the focus of the discussion was not related to the research. For example, Deep Residual Network (ResNet) was mentioned in the related research but was not followed up in subsequent sections, making the introduction to ResNet not relevant to this research.

Experimental design

Methods
• Examples of shallow and deep features related to service quality in this research need to be explained in more detail.

Datasets
• Data is only from one company, which is very limited in size and difficult to be generalized.
• It was mentioned that data was collected from the survey, but no explanation was made on how the survey was conducted.
• The size of the data such as the number of respondents was missing.
• If it is cross-border, the data from foreign consumers need to be collected. However, the authors did not mention consumers from the USA.
• Authors did not mention if they have made some pre-processing on the data other than descriptive statistics.
• There was no discussion about the 16 indicators used in the research.

Validity of the findings

• The experimental training and testing process were omitted.
• There was no comparative evaluation with any benchmark methods to prove the effectiveness of the proposed methods.

Reviewer 2 ·

Basic reporting

The quality of the submission is fine; of course, some parts need to improve.

Experimental design

The appendix should be moved to the result section to show the results of the study clearly. In total, this section is presented in well structure and presentation.

Validity of the findings

Needs to work on the conclusion to improve; I provided a comment for this section.

Additional comments

- The abstract lacks usefulness; I recommend that the authors revise this section by incorporating the study background, research objectives, research method, results, and conclusion.

- The study lack of state-of-the-art literature. In section two, the authors should discuss relevant published papers, emphasizing the study's contributions and novelty.

- The introduction section should highlight both the research method and the study's novelty.

- To enhance readability, it is advisable to relocate all tables and figures to the body of the paper.

- Section two would benefit from a brief discussion at its conclusion.

- The presentation of equations should be improved for better clarity.

- The research method section requires an introduction to provide context.

- Consider adding a diagram or framework in the paper to illustrate the proposed method and research procedure.

- The conclusion should be expanded and enhanced, addressing study limitations and offering recommendations for future work.

- I recommend that the authors consider sending their paper to a proofreader for further polishing.

Reviewer 3 ·

Basic reporting

A novel fast depth perception network is proposed. The network extracts a set of shallow features and deep features from the pre-training model trained on DCNN, and then inputs the two sets of features into DPM to obtain a set of depth intensive vectors that can represent semantic information. Taking a cross-border e-commerce enterprise as an example, this paper explores and selects the logistics service index of the enterprise, designs the evaluation questionnaire of the logistics service of a cross-border e-commerce, and uses the analytic hierarchy process to analyze the logistics service index closely related to customer satisfaction. In my opinion, the topic is very interesting, and can be accepted with a revision.

1 Backpropagation is carried out according to the set loss function, and the final result is obtained through softmax after the termination condition is reached. The aspects of using multi-layer perceptrons that need attention should be explained.
2 The variance is maximized when u1 is set to the eigenvector with the largest eigenvalue, and this eigenvector becomes the first principal component. The advantages of PCA technology should be analyzed.
3 The biggest application of transfer learning to deep learning and computer vision is fine-tuning. Please explain the fine tuning process.
4 Figure 1 shows the construction of the multi-layer perceptron. However, the text in the picture should not be bold.
5 Some variables in this article are not in italics, such as "Cronbach's α".
6 This paper puts forward the strategy of averaging near scale. Its necessity needs to be explained.
7 The three steps of multi-angle perception convolutional neural network can be explained.
8 General backpropagation algorithms based on gradient descent need to pre-consider such as learning rate, momentum term, number of iterations, etc., and the selection of these hyperparameters requires a large number of experiments to make choices. Describe the situation in the BLS algorithm.

Experimental design

none

Validity of the findings

none

Additional comments

none

Reviewer 4 ·

Basic reporting

This paper introduces a framework used to assess the service quality in cross-border e-commerce. The framework relies on transfer learning using multi-angle perception and convolutional neural networks. The paper is well-written, clear, and employs professional language. However, I believe that some structural improvements are necessary to enhance its readability. For instance, I occasionally found it challenging to follow the flow of ideas. For instance, in the introduction, the authors begin with a description of the significance of service quality in current competition, then provide an overview of deep learning techniques, and subsequently return to the description of e-commerce in China. I suggest it might be more effective to present the importance of e-commerce in China before delving into the details of deep learning techniques. Furthermore, I propose consolidating subsections 3-1, 3-2, and 3.3 into a separate section to provide background information before delving into the specific contributions.
The bibliography contains recent and relevant references. However, I recommend a thorough review of this section or using reference management tools (e.g., reference 8 is duplicated as 21). Additionally, the authors mark each reference with [J], perhaps to identify journals, but this practice deviates from standard bibliographic reference formatting.
I also suggest adding a comparative study at the end of the related work section to highlight knowledge gaps. In a review, it's insufficient to merely mention articles addressing the same issue without analyzing and comparing them. Regrettably, the paper is not entirely self-contained, as several abbreviations are not explicitly defined (e.g., PCA at line 234, BLS at line 134, Cronbach's alpha at line 336, and ZQ at line 367).

Experimental design

The issue addressed in this study is significant. Given the competition among businesses in the e-commerce sector, it is crucial to identify the service quality criteria that impact customer choices. However, I could not ascertain the specific definition of quality upon which the authors base their study.
While the question is well-framed, the authors state that they propose three contributions (lines 56 -63). Personally, I believe there is a single contribution comprised of three steps.
The paper aims to introduce several criteria, and the proposed model provides the most relevant ones. Firstly, the authors do not clarify the input criteria. Upon reviewing the attached data, I noticed that the data inputs include (Order number, Order bank, Sales time, Delivery time, Delivery status of SKU, SKU, User feedback, Area, Number, Price). Are these the criteria used as inputs? These data do not represent multiple criteria requiring a distinction between relevant and irrelevant criteria. Furthermore, in the results section, the authors mention five dimensions and 14 output indicators, but they only refer to three dimensions in the correlation analysis and four dimensions in the results. I am unsure about the fifth dimension. Additionally, the authors use the terms "safety" and "security" interchangeably (lines 325 - 357). I believe that these two concepts have minor differences according to specialized literature and information security standards.
The data associated with the paper lack clarity. It would be helpful to organize them into columns and provide a description for each column. I recommend that the authors provide a more detailed description of their methodology. In line 226, the authors mention a second transfer learning method without any explanation. It is essential to specify the type of transfer learning to which the proposed method belongs.

Validity of the findings

The contribution presented in this paper is significant, novel, and original. The authors provide both data and code for their research. However, I suggest that the authors enhance the "future works" section by providing more detailed information.

---

## Round 0.2 · Minor Revisions

As per comments from original reviewers, a minor revision decision can be recommended.

Reviewer 1 ·

Basic reporting

Abstract:
Training and testing is a methodology that must be included in research related to experiments in the machine learning domain. Removing training and testing is not what is expected from the authors.

Literature Review
There is no citation given to support CNN. Please add citations related to the discussion related to CNN.

Experimental design

No comment

Validity of the findings

The previous comment: 'There was no comparative evaluation with any benchmark methods to prove the effectiveness of the proposed methods.'
This comment has not been addressed. Although the authors respond to this comment in the rebuttal table, the changes were not found in the text.

Reviewer 3 ·

Basic reporting

no comment

Experimental design

no comment

Validity of the findings

no comment

Additional comments

no comment

Reviewer 4 ·

Basic reporting

This paper presents a framework designed for evaluating service quality in cross-border e-commerce. The framework leverages transfer learning through multi-angle perception and convolutional neural networks. The paper is skillfully written, clear, and employs professional language throughout. Following the introduction, the authors provide an overview of related work, incorporating various recent and pertinent papers. In the methodology section, they detail the primary methods employed in their approach to transfer learning, such as deep convolutional neural networks and a depth perception module. Subsequently, the paper presents the results obtained from their methodology. I recommend enhancing the related work section by including a more explicit comparative study.

Experimental design

The matter explored in this study holds considerable significance. In the competitive landscape of the e-commerce sector, identifying the service quality criteria influencing customer choices is crucial. The research question is well-articulated, and the chosen method (transfer learning) is justified. The related work is thoroughly examined.

Validity of the findings

The contribution presented in this paper is significant, novel, and original. The authors provide both data and code for their research.

---

## Round 0.3 · accepted · Accept

All previous reviewers have no comments, and this revised paper can be accepted for publication.

Reviewer 1 ·

Basic reporting

No comment.

Experimental design

No comment.

Validity of the findings

The author(s) claimed that there is no existing work that can be used as the benchmark. They however will consider benchmarking in future research.

Additional comments

No comment.

Reviewer 2 ·

Basic reporting

The paper is improved well.

Experimental design

Improved

Validity of the findings

Improved

Additional comments

Improved

Reviewer 3 ·

Basic reporting

No comment

Experimental design

No comment

Validity of the findings

no comment

Additional comments

no comment